# An Area Coverage and Energy Consumption Optimization Approach Based on Improved Adaptive Particle Swarm Optimization for Directional Sensor Networks

**DOI:** 10.3390/s19051192

**Published:** 2019-03-08

**Authors:** Song Peng, Yonghua Xiong

**Affiliations:** 1School of Automation, China University of Geosciences, Wuhan 430074, China; pengsong0916@163.com; 2Hubei Key Laboratory of Advanced Control and Intelligent Automation for Complex Systems, Wuhan 430074, China

**Keywords:** directional sensor network, area coverage, cluster, particle swarm optimization, energy consumption balance

## Abstract

Coverage is a vital indicator which reflects the performance of directional sensor networks (DSNs). The random deployment of directional sensor nodes will lead to many covergae blind areas and overlapping areas. Besides, the premature death of nodes will also directly affect the service quality of network due to limited energy. To address these problems, this paper proposes a new area coverage and energy consumption optimization approach based on improved adaptive particle swarm optimization (IAPSO). For area coverage problem, we set up a multi-objective optimization model in order to improve coverage ratio and reduce redundancy ratio by sensing direction rotation. For energy consumption optimization, we make energy consumption evenly distribute on each sensor node by clustering network. We set up a cluster head selection optimization model which considers the total residual energy ratio and energy consumption balance degree of cluster head candidates. We also propose a cluster formation algorithm in which member nodes choose their cluster heads by weight function. We next utilize an IAPSO to solve two optimization models to achieve high coverage ratio, low redundancy ratio and energy consumption balance. Extensive simulation results demonstrate the our proposed approach performs better than other ones.

## 1. Introduction

Directional sensor networks (DSNs) consist of low-energy, low-cost, small-size and multi-function directional sensor nodes which cooperate to collect video monitoring data of target area and communicate with a remote base station (BS) [1]. DSNs have wide application prospects in industry control, environment monitoring and city management etc. [2]. Coverage is an important factor which directly reflects the service quality of DSN. In actual application scenarios, random deployment of sensor nodes will lead to many coverage overlapping areas and coverage blind areas. Besides, the directional sensor nodes are generally powered by battery with limited energy, the premature death of nodes will also affect the service quality of network. Therefore, in a random deployed DSN with limited energy, we should not only consider how to reduce coverage overlapping areas and coverage blind areas but also consider how to prolong network lifetime by balancing energy consumption of each node.

In a randomly deployed DSN, the directional sensor nodes are rotatable and with limited energy. Considering that the position, sensing radius and sensing angle of sensor nodes are invariable, it is necessary to improve coverage ratio and reduce redundancy ratio by rotating sensing direction. Traditional coverage optimization methods generally consider that coverage ratio and redundancy ratio have a negative correlation. The redundancy ratio will certainly drop with the improvement of coverage ratio. However, the correlation between coverage ratio and redundancy ratio is not obvious in the application scenario where the target area is multiply covered. If the coverage ratio is improved, the redundancy ratio may also improve if the new covered area has redundancy. On the contrary, with decrement of coverage ratio, the redundancy ratio may also get reduced if reduced coverage area has redundancy. To address the problems of multiple area coverage, this paper proposes a sensing direction rotation approach to achieve high coverage ratio and low redundancy ratio by searching for optimal sensing direction group. This problem is essentially a combinatorial optimization problem which is NP-hard [3].

For a directional sensor node, it is generally believed that the energy consumption consists of two main components: the sensing energy consumption for data collection and the communication energy consumption for data transmission. Considering that all the sensor nodes in a monitoring area are opened during the service period, it is not necessary to consider how to control the sensing energy consumption of the nodes by turning the nodes on or off. In order to achieve energy consumption balance of the network, this paper only considers how to control the communication energy consumption.

Communication energy consumption mainly comes from data communication between nodes, and it is important to reduce redundant data communication by optimizing the topology of the network and designing a reasonable data communication protocol. Cluster is one of the most effective network layering optimization technologies which is employed to conserve energy of sensor nodes [4]. The structure of a clustered DSN is shown in Figure 1.

The network is divided into many clusters, and each cluster consists of a cluster head and multiple member nodes. The member node is only responsible for collecting and transmitting data. The cluster head receives monitoring data from member nodes within cluster and send them to a remote BS after fusion in addition to having the same function as the member node. In the process of clustering, cluster head selection plays a vital role in distributing communication energy consumption on each node as it has a direct effect on the energy conservation of member nodes. This paper mainly considers the total residual energy ratio and energy consumption balance degree of the cluster head candidate nodes. Compared to the traditional cluster head selection approach which can only be applied to small-scale networks, the proposed approach can effectively achieve energy consumption balance for networks of all scales. The cluster head selection is also an optimization problem which has been proven to be NP-hard [5].

Based on the above analysis, the sensing direction rotation and cluster head selection are all NP-hard problems. For this type of problems, a meta-heuristic algorithm such as simulated anneal, genetic algorithm and particle swarm optimization are suitable for solving it [6]. The designed algorithm needs to consider time, resource consumption and accuracy. Particle swarm optimization (PSO) is a better choice for the NP-hard problems due to its easy implementation and high precision. In order to guarantee the adaptability of the algorithm, we make an improvement for the inertia weight.

In this paper, we mainly concentrate on area coverage optimization and energy consumption optimization problems. For the area coverage problem, we set up a multi-objective optimization model due to the uncertainty between coverage ratio and redundancy ratio. We reduce coverage blind areas and coverage overlapping areas by sensing direction rotation. For energy consumption optimization problem, we set up cluster head selection optimization model in order to achieve communication energy consumption balance. We make the energy consumption evenly distribute on each sensor node by considering the total residual energy ratio and energy consumption balance degree of the cluster head candidates nodes in each round. We also propose a cluster formation algorithm in which the member nodes join their cluster heads by energy efficiency weight function. We next propose an improved adaptive particle swarm optimization (IAPSO) to solve the two optimization models to achieve high coverage ratio, low redundancy ratio and energy consumption balance. Compared to traditional PSO, we make an improvement for inertial weight, and the IAPSO has higher convergence ratio and operator precision.

The rest of the paper is organized as follows. Section 2 reviews the previous related work. The preliminaries of network model, directional sensor model and energy consumption model are provided in Section 3. In Section 4, the multi-objective area coverage optimization problem is described. Section 5 analyzes the cluster-based energy consumption optimization problem. Section 6 presents our proposed approach. Section 7 conducts simulation experiment to evaluate the efficiency of the proposed approach. The conclusion is in Section 8.

## 2. Related Works

In order to improve service quality of DSN, it is necessary to consider coverage and energy consumption. On one hand, we should study how to improve area coverage quality by rotating sensing directions of sensor nodes, on the other hand, we should consider how to guarantee energy consumption balance by clustering network. This section reviews the existing area coverage optimization approaches and cluster-based energy consumption optimization approaches.

### 2.1. Area Coverage Optimization Approaches

Many optimization approaches have been proposed for area coverage problems. Among them, some explore the movement-assisted sensor node deployment by using mobile nodes to improve coverage quality [7,8,9,10,11,12], and these approaches are only suitable to the scenario where the sensor node is omnidirectional.

In Ref. [7], the authors propose a virtual force algorithm as a sensor deployment strategy to enhance the coverage after an initial random placement of sensor nodes. In Ref. [8], a localized scan-based movement-assisted sensor deployment method is proposed to use scan and dimension exchange to achieve a balanced state. To address another coverage problem, authors in Ref. [9] propose the optimal coverage in directional sensor networks problem to cover maximal area while activating as few as sensor nodes as possible. To study how to place mobile sensor nodes to get high coverage, Ref. [10] designs two sets of distributed approaches for controlling the movement of sensor nodes. In Ref. [11], the authors propose a harmony search optimization algorithm to solve area coverage problem. A previous study in Ref. [12] proposes a learning automata-based heuristic algorithm to find a near optimal solution to the proxy equivalent degree-constrained minimum-weight extension of the connected dominating set problem.

With the increasing application of surveillance camera and video sensing network, some approaches for directional sensor networks have been proposed. Some consider both the rotation and movement of sensor nodes [13,14,15,16,17,18], and the others only consider the rotation of static sensor nodes [19,20,21,22].

Ref. [13] proposes a moving algorithm based on virtual forces of directional sensor nodes to eliminate coverage holes. In Ref. [14], the authors present distributed self-deployment schemes of mobile sensor node according to the circumcenter and incenter of sensing direction. Taking another perspective on the coverage problem, authors in Ref. [15] propose two Enahanced Deployment algorithms namly EDA-1 and EDA-2 to maximize coverage ratio for heterogeneous directional mobile network. To address the deployment problem, authors in Refs. [16,17] transform the area coverage problem into cell coverage problems by exploiting the Voronoi diagram. Ref. [16] proposes a distributed approach to enhance the overall area coverage, and Ref. [17] presents three local coverage optimization algorithm to improve coverage ratio. Different from above two approaches, Ref. [18] presents several coverage increment algorithms namely vertex-based adjustment with Voronoi diagram (V-VD), edge-based adjustment with Voronoi diagram (E-VD), edge-based adjustment with Delaunay triangulation (E-DT) and angle-based adjustment with Delaunay triangulation (A-DT).

Authors in Ref. [19] introduce the concept of sensing centroid into an omni-directional one, and the artificial fish-swarm algorithm is utilized to reduce the coverage hole and achieve the global coverage optimization. In Ref. [20], the proposed approach studies how many sensor nodes are needed to meet a given required coverage probability under the circumstance that the sensing radius is adjustable. However, these two approaches ignore the redundancy ratio which is also a vital factor for improving coverage quality. In Refs. [21,22], the coverage ratio and redundancy ratio are considered at the same time. Among those, Ref. [21] proposes a virtual potential field based coverage algorithm to increase the coverage ratio by forcing sensor nodes to turn form overlapping region to coverage holes after random deployment. In Ref. [22], a virtual centripetal force-based coverage-enhancing algorithm is proposed to enhance the coverage by redeploying sensor nodes under the repel force-based centripetal force to shut off redundant nodes. However, these two approaches consider that coverage ratio and redundancy ratio have a negative correlation, and they are not suitable to the scenario where the target area is multiply covered.

Our proposed approach addresses the area coverage problem of DSN consisting of static and rotatable sensor nodes. With the objective of improving coverage ratio and reducing redundancy ratio by sensing direction rotation, we set up a multi-objective model in order to achieve high coverage ratio and low redundancy ratio. We next utilize an IAPSO to search for optimal sensing direction group to achieve multi-objective optimization for DSN. Compared to existing approaches [21,22], our proposed area coverage optimization approach can be suitable for multiply covered area where the negative correlation between coverage ratio and redundancy ratio is not obvious, and the IAPSO can effectively avoid local optima to some degree.

### 2.2. Cluster-Based Energy Consumption Optimization Approaches

A large number of cluster-based energy consumption optimization approaches have been proposed for energy consumption balance, and they can be categorized into two types: meta-heuristic approaches and nature-inspired approaches.

Meta-heuristic approaches are proposed on the basis of intuitive or empirical construction, and they usually have some randomness and uncertainty. The classic LEACH (Low-Energy Adaptive Clustering Hierarchy) is proposed in Ref. [23]. LEACH can effectively reduce energy consumption to some extent, but it may select a cluster head with low residual energy and shorten the network lifetime. Therefore, many approaches have made some improvement on the basis of LEACH. HEED in Ref. [24] selects cluster heads according to a hybrid of the node residual energy and a secondary parameter such as node proximity to its neighbors or node degree. PEGASIS in Ref. [25] makes sensor node communicate only with a close neighbor and takes turns transmitting to the base station. In Ref. [26], TL-LEACH saves energy consumption by using random rotation of local cluster base station. DL-LEACH in Ref. [27] increases the energy efficiency of sensor node by reducing the transmission distance and simplifying the transmission routine for short-range transmission. In Ref. [28,29,30], the authors propose energy efficient cluster based routing schemes for reliable networks. In Ref. [31], E-LEACH uses homomorphic encryption to provide secure data aggregation and reduce energy consumption. According to Ref. [32], M-LEACH is better than the TL-LEACH, and E-LEACH is better than M-LEAC. Although these approaches have better performance than LEACH, they cannot guarantee the energy consumption balance of sensor nodes when the size of the network is large.

Other approaches are proposed in the literature based on application of nature-inspired approaches. Among them, some are the improved LEACH. For example, LEACH-C in Ref. [33] has better performance than LEACH because it considers the intra-cluster distance and residual energy of sensor nodes in the phase of cluster head selection. In Ref. [34], a PSO based approach is proposed to select the optimal location of cluster head. Although these two approaches can guarantee the energy consumption balance of cluster head, they ignore the sink distance which is also an important factor to improve energy efficiency for direct communication of data to the base station. LEACH-FL takes battery level, distance and node density into consideration [35], and it easily raises complexity and the accuracy problem in the fuzzification and defuzzification process.

As nature-inspired approaches which only utilize approximation algorithm to achieve energy consumption balance. PSO is utilized to achieve cluster head selection in Refs. [36,37,38]. The authors in Ref. [36] increase the network lifetime by reducing the total communication distance, and the authors in Ref. [37] consider the intra-cluster distance and the residual energy of cluster head candidates. In Ref. [38], the proposed approach considers the residual energy, intra-cluster distance, node degree and head count of the probable cluster heads. However, these three approaches cannot guarantee the performance with the size of the network verifying because they ignore the sink distance. In Ref. [39], the authors propose an energy efficient clustering scheme based on recent variable population based chemo-inspired approach. To prolong the network lifetime, a PSO-based multiple-sink placement algorithm is proposed in Ref. [40]. However, it ignores the fault-tolerance of a network.

Our proposed cluster-based approach belongs to a nature-inspired approaches. In the phase of cluster head selection, we consider the total residual energy and energy balance degree of the cluster head candidate nodes, and the two parameters can effectively guarantee the energy consumption balance of network whereas the existing approaches only consider the distance parameters or residual energy [33,34,36,37,38]. We also propose a cluster formation algorithm in which the member nodes join in a cluster head by a weight function. However, in the existing approaches [23,33,36,37,38], the member nodes join the cluster head by only considering distance, and it may cause imbalance load of cluster head energy consumption.

Based on the above analysis, area coverage optimization and energy consumption optimization are two key processes for improving service quality of DSN. However, they are usually studied in separate according to existing literatures. Most of the exiting approaches about area coverage optimization for DSN only focus on how to improve coverage ratio and ignore how to reducing redundancy ratio. Although some researchers consider the redundancy ratio, their proposed approaches are only suitable to the scenarios where the coverage ratio and redundancy ratio have the negative correlation. The existing cluster-based energy consumption optimization approaches cannot effectively guarantee energy consumption balance for network with different size because they usually only consider distance parameters and residual energy in the cluster head selection phase. In this paper, we take both area coverage quality and energy consumption into account and propose an approach based on IAPSO to achieve high coverage ratio and low redundancy ratio and energy consumption balance for DSN.

The contributions of this paper are as follows:We propose a multi-objective area coverage optimization model which considers coverage ratio and redundancy ratio in order to reduce coverage blind areas and coverage redundant areas. This model is suitable for the scenario where the target area is multiply covered.We propose a cluster head selection optimization model which considers the total residual energy ratio and energy balance degree of the cluster head candidate nodes to guarantee energy efficiency. We also propose an energy efficiency algorithm in the cluster formation phase.We utilize an improved adaptive particle swarm optimization (IAPSO) to solve multi-objective area coverage optimization model and cluster head selection optimization model to achieve high coverage ratio, low redundancy ratio and energy consumption balance. Compared to traditional PSO, IAPSO has higher convergence ratio and operator precision.

## 3. Preliminaries

### 3.1. Network Model

Suppose that there are *n* sensor nodes randomly deployed in the $m_{1}\times m_{2}$ rectangle monitoring area. In order to simplify the problem analysis, some assumptions about the DSN scenario are given as follows.
(1)Each sensor node could collect data and send them to BS.(2)The position information of sensor nodes could be obtained by BS.(3)All directional sensor nodes had the same initial energy, sensing radius, angle of view (AoV) and communication ability.(4)Each sensor node could be cluster head or member node.(5)Each sensor node could reduce data transmission by data fusion.(6)All directional sensor nodes could guarantee the network connectivity.

### 3.2. Directional Sensing Model

Compared to traditional omni-directional sensor node, which has a circle sensing range, the sensing area of a directional sensor node is a smaller sector-like area [16]. As shown in Figure 2, the sensing area of a directional sensor node is a sector determined by a 5-tuple $\left(x_{s},y_{s},\vec{D},\alpha ,R\right)$, where $x_{s},y_{s}$ is the coordinate of the node, $\vec{D}$ is the working direction of *S*, α is the angle of view (AoV), and *R* is the sensing radius. In addition, ρ is the angle value of working direction $\vec{D}$ relative to the horizontal.

Figure 2 also shows the coverage situation of a point in the monitoring area, it can be observed that point *P* can be covered by the directional sensor node *S* if and only if the following two conditions are satisfied.
(1)Let $dis\left(S,P\right)$ be the distance between $S\left(x_{s},y_{s}\right)$ and $P\left(x,y\right)$, and it must be no more than the sensing radius *R*, i.e., $dis\left(S,P\right)\leq R$:
(1)$$dis\left(S,P\right)=\sqrt{{\left(x-x_{s}\right)}^{2}+{\left(y-y_{s}\right)}^{2}}\leq R$$(2)The absolute included angle between $\vec{SP}$ and working direction $\vec{S}$ must be no more than the half of AoV, i.e., α/2
(2)$$\delta =\arccos\frac{\vec{SP}\cdot \vec{D}}{\left|\vec{SP}\right|}\leq \frac{\alpha }{2}$$

In brief, a point $P\left(x,y\right)$ is said to be covered by the sensor node *S* if and only if (1) and (2) are satisfied with the sensor node coordinate $\left(x_{s},y_{s}\right)$ and characteristic parameters *R* and α.

### 3.3. Energy Consumption Model

The energy consumption model in this paper is the same as the radio model in Ref. [23]. In this model, the transmitter dissipates energy to run the radio electronics and the power amplifier. The receiver dissipates energy to run the radio electronics.

The radio electronics and power amplifier are responsible to send data. The energy consumption includes two modes: (1) free space mode and (2) multipath fading mode. If the data transmission distance *d* is less than the threshold distance *d*${}_{0}$, free space mode is used. Otherwise multipath fading mode is used. Let the Signal to Noise Ratio (SNR) be reasonable in the circuit, the energy consumption that a sensor node sends *l*-bit data packet is given by the following equations.
(3)$$E_{Tx}\left(l,d\right)=\left\{\begin{array}{c} l\times E_{elec}+l\times \varepsilon_{fs}\times d^{2},d\leq d_{0}\\ l\times E_{elec}+l\times \varepsilon_{mp}\times d^{4},d>d_{0} \end{array}\right.$$where $E_{elec}$ is the energy dissipated per bit to run the transmitter or the receiver circuit. $\varepsilon_{fs}$ and $\varepsilon_{mp}$ are the power amplification energy consumption coefficients in different energy consumption modes. $d_{0}$ is the threshold transmission distance, and $d_{0}$ is given by
(4)$$d_{0}=\sqrt{\frac{\varepsilon_{fs}}{\varepsilon_{mp}}}$$

To receive *l*-bit data, the energy consumed by the receiver is given by
(5)$$E_{Rx}\left(l\right)=l\times E_{elec}$$

In general, the communication energy parameters can be set as: $E_{Tx}=50$ nJ/bit, $\varepsilon_{fs}=10\;\mathrm{pJ}/\mathrm{bit}/m^{2}$, $\varepsilon_{mp}=0.0013\;\mathrm{pJ}/\mathrm{bit}/m^{4}$, $d_{0}=87$ m.

## 4. Multi-Objective Area Coverage Optimization Problem

To improve the coverage quality of DSN, we should improve coverage ratio and reduce redundancy ratio by sensing direction rotation, and it is a multi-objective optimization problem. It is important to derive a multi-objective area coverage optimization model after coverage situation verification.

### 4.1. Coverage Situation Verification

In order to calculate coverage ratio and redundancy ratio, we conduct grid partition in the monitoring area. As shown in Figure 3a, the area can be regarded as a collection of points when the grid is small enough. Let the monitoring area be a $m_{1}\times m_{2}$ rectangle area, and $P\left(x,y\right),0\leq x\leq m_{1},0\leq y\leq m_{2}$ is a point in the area. We define 2-dimensional arrays *C* and *R* to represent coverage and redundancy situations of each point. We justify their coverage and redundancy situations according to Equations (Equation 1) and (Equation 2). If the point can be covered by sensor nodes, $C\left[x\right]\left[y\right]=1$, otherwise $C\left[x\right]\left[y\right]=0$. If the point can be covered by at least two sensor nodes, $R\left[x\right]\left[y\right]=1$, otherwise $R\left[x\right]\left[y\right]=0$.

Due to the fixed position and limited sensing range of directional sensor node, we only need to justify the coverage and redundancy situations of the grid points in the rectangle area in the Figure 3b in order to reduce calculation complexity, and the four peak points are defined as follows.
(6)$$x_{sl}=\max\left\{x_{s}-R,0\right\}$$
(7)$$x_{sh}=\min\left\{m_{1},x_{s}+R\right\}$$
(8)$$y_{sl}=\max\left\{y_{s}-R,0\right\}$$
(9)$$y_{sh}=\min\left\{y_{s}+R,m_{2}\right\}$$

In Algorithm 1, the four peak coordinates are determined according to the coordinate of sensor node $\left(x_{s},y_{s}\right)$ and sensing radius *R*. The algorithm justifies the coverage and redundancy situations of each point in the rectangle area, and the values of arrays *C* and *R* are determined. The circle number of the algorithm is *n*, and the time complexity is only $o\left(R^{2}\right)$ when the coverage situation of each point in the rectangle area is justified.

**Algorithm 1:** Coverage verification algorithm based on the sensing area of sensor node**Input:** Sensor nodes group: $s=\left\{s_{1},s_{2},\ldots s_{n}\right\}$   Sensing direction and sensing radius of sensor node: $\vec{D},R$   The length and width of target area: $m_{1},m_{2}$**Output:** The coverage situations of grid points in target area1: Calculate the coordinates of four peak points $x_{sl},x_{sh},y_{sl},y_{sh}$2: **for**
i=1
to
*n*
**do**3:        **for**
$x=x_{sl}$
to
$x_{sh}$
**do**4:               **for**
$y=y_{sl}$
to
$y_{sh}$
**do**5:                     **if**
$P\left(x,y\right)$ can be covered by $S\left(x_{s},y_{s}\right)$
**then**6:                          $C\left[x\right]\left[y\right]=1$;%Justify whether $P\left(x,y\right)$ is covered.7:                     **end if**8:                     **if**
$P\left(x,y\right)$ can be covered by at least two sensor nodes. **then**9:                          $R\left[x\right]\left[y\right]=1$;%Justify whether $P\left(x,y\right)$ is covered by at least two sensor nodes.10:                   **end if**11:              **end for**12:       **end for**13: **end for**

### 4.2. Multi-Objective Area Coverage Optimization Model

Based on coverage situation verification for grid points, we set up a multi-objective area coverage optimization model in order to improve coverage quality. The relative fitness function considers the following two factors.

(1) Missed ratio 

After grid partition is conducted in the area, coverage ratio is approximately equal to the ratio of the sum of covered grid points to the sum of points in the monitoring area. Coverage ratio is an important evaluating factor for monitoring quality, and it is important to reduce the missed coverage ratio. One of the objective function is as follows.
(10)$$f_{1}=1-\frac{\sum_{x=0}^{m_{1}}\sum_{y=0}^{m_{2}}C\left[x\right]\left[y\right]}{\left(m_{1}+1\right)\left(m_{2}+1\right)}$$where $\sum_{x=0}^{m_{1}}\sum_{y=0}^{m_{2}}C\left[x\right]\left[y\right]$ is the sum of the points covered by one directional sensor node, and $\left(m_{1}+1\right)\left(m_{2}+1\right)$ is the sum of grid points in monitoring area. The objective is to minimize the objective function value.

(2) Redundancy ratio 

It is defined as the ratio of the sum of grid points covered by at least two sensor nodes to the sum of points in the monitoring area. Considering the deployment cost, it is wise to reduce redundancy ratio to avoid the waste of resources. The other objective function is defined as follows.
(11)$$f_{2}=\frac{\sum_{x=0}^{m_{1}}\sum_{y=0}^{m_{2}}R\left[x\right]\left[y\right]}{\left(m_{1}+1\right)\times \left(m_{2}+1\right)}$$where $\sum_{x=0}^{m_{1}}\sum_{y=0}^{m_{2}}R\left[x\right]\left[y\right]$ is the sum of the points covered by at least two directional sensor nodes, and $\left(m_{1}+1\right)\times \left(m_{2}+1\right)$ is the sum of grid points in monitoring area. The objective is to minimize the objective function value.

In our approach, we choose to minimize the linear combination of the above two objective functions instead of minimizing them individually because the two objectives conflict with each other. To optimize the two objectives at the same time, the coverage optimization fitness function is defined as follows.
(12)$$Fitness1=\alpha \times f_{1}+\left(1-\alpha \right)\times f_{2},0<\alpha <1$$where the α is a constant defined by weighing the importance of the sub-objective, and its value is between 0 and 1. Our objective is to minimize the fitness value in order to achieve better area coverage optimization effect.

The multi-objective area coverage problem can be summarized as searching for a sensing direction group of sensor nodes $\left(\vec{D_{1}},\vec{D_{2}},\vec{D_{3}},\vec{D_{4}},\ldots ,\vec{D_{n}}\right)$. When all sensor nodes in network rotate their sensing directions according to this group, we can improve coverage ratio and reduce redundancy ratio.

## 5. Cluster-Based Energy Consumption Optimization Problem

The main process of cluster consists of two phases: cluster head selection and cluster formation. In the cluster head selection phase, we select the optimal cluster head group amongst the directional sensor nodes in monitoring area. In the cluster formation phase, the member nodes choose their cluster heads to join in by energy efficiency weight. This section designs a cluster head selection optimization model and propose a cluster formation algorithm based on cluster head weight.

### 5.1. Cluster Head Selection Optimization Model

To address the cluster head selection problem, it is important to achieve energy consumption balance of network. We set up a cluster head selection optimization model which depends on the following two relative energy efficiency factors.

(1) Total residual energy ratio of cluster head candidates 

It is defined as the ratio of total residual energy of all active sensor nodes $n_{i},i=1,2,\ldots ,n$ in the DSN to the total residual energy of cluster head candidates $CH_{c},c=1,2,\ldots ,k$ in current round. In the data transmission phase, the cluster heads receive the data from member nodes and send aggregated data to the BS. The energy consumption of cluster heads is much more than member nodes so it is necessary to guarantee enough residual energy. One of the objective function is defined as follows.
(13)$$f_{3}=\sum_{i=1}^{n}S\_E\left(n_{i}\right)/\sum_{c=1}^{k}C\_E\left(CH_{c}\right)$$where $S\_E\left(n_{i}\right)$ is the residual energy of sensor node $n_{i}$, and $C\_E\left(CH_{c}\right)$ is the residual energy of cluster head candidate $CH_{c}$ in current round. The objective is to minimize the objective function value.

(2) Energy consumption balance degree of the cluster head candidates 

It is defined as the energy consumption variance of the cluster head candidates in current round. To improve the lifetime of network, the BS needs to ensure that the energy consumption is evenly distributed among all the cluster heads so it is necessary to guarantee all the cluster heads consume approximate energy. The second objective function is defined as follows.
(14)$$f_{4}=\sqrt{{\sum_{c=1}^{k}\left[CE_{r}\left(CH_{c}\right)-u\right]}^{2}}$$where $CE_{r}\left(CH_{c}\right)$ is the energy consumption of a cluster head candidates in current round, and *u* is the average energy consumption of all cluster head candidates. The objective is to minimize the objective function value.

The above two objectives are not conflicting with each other so the cluster head selection fitness function is defined as a linear combination of the two objective functions.
(15)$$Fitness2=\beta \times f_{3}+\left(1-\beta \right)\times f_{4},0<\beta <1$$where the β is a constant defined by weighing the importance of the sub-objective, and its value is between 0 and 1. Our objective is to minimize the fitness value in order to achieve better cluster head selection optimization effect.

The cluster head selection optimization problem can be summarized as searching for a cluster head group $\left(CH_{1},CH_{2},CH_{3},CH_{4},\ldots ,CH_{c}\right)$ to minimize the fitness value. When the fitness is minimal, we can effectively guarantee energy consumption balance.

### 5.2. Cluster Formation

For the cluster formation process, we should balance the load of cluster head energy consumption. The cluster formation depends on the following two parameters.

(1) The total distance of data transmission

In order to consume less energy, the total distance of data transmission should be as small as can e acheived. The distance includes two aspects i.e., the inner-cluster distance and sink distance. The inner-cluster distance is the distance between the member node $S_{i},i=1,2,\ldots ,n-k$ and cluster head $CH_{c},c=1,2,\ldots ,k$, and the sink distance is the distance between the cluster head and BS. The total distance is given by the following equation.
(16)$$total\_dis\left(S_{i}\right)=dis\left(S_{i},CH_{j}\right)+dis\left(CH_{j},BS\right)$$where $dis\left(S_{i},CH_{j}\right)$ is the inner-cluster distance, and $dis\left(CH_{j},BS\right)$ is the sink distance. BS denotes the base station. The total distance of transmission is the sum of inner-distance and sink distance.

(2) The residual energy of cluster head 

In the phase of data transmission, the cluster heads receive data from member nodes and send them to BS after fusion. In general, the energy consumption of cluster head is much higher than member nodes so it is important to guarantee the sufficient residual energy of cluster head. Therefore, a member node should join a cluster head which has a higher residual energy i.e., $C\_E\left(CH_{j}\right)$.

To take the above two factors into account at the same time, a weight function of cluster head is given as follows.
(17)$$C\_W\left(S_{i},CH_{j}\right)=\gamma \times \frac{C\_E\left(CH_{j}\right)}{total\_d\left(S_{i}\right)}$$where $C\_W\left(S_{i},CH_{j}\right)$ is a linear combination of the total distance of data transmission and the residual of energy of cluster head, and γ is a constant defined by user.

Algorithm 2 shows the process of cluster formation by using cluster weight. The member nodes in the monitoring area select the cluster head of lowest weight to join in.

**Algorithm 2:** Cluster formation algorithm based on cluster head weight**Input:** Sensor nodes group: $s=\left\{s_{1},s_{2},\ldots ,s_{n}\right\}$   Cluster head group: $CH=\left\{CH_{1},CH_{2},CH_{3},\ldots ,CH_{k}\right\}$   The residual energy of all sensor nodes: S_E   The residual energy of cluster heads: C_E**Output:** The ids of cluster heads the member nodes join: S_clusterhead1: **for**
i=1
to
*n*
**do**2:       **if**
S_E>0
**then**3:           Calculate $C\_W\left(S_{i},CH_{1}\right)$4:           $temp=C\_W\left(S_{i},CH_{1}\right)$5:           $S\_clusterhead\left(i\right)=1$6:           **for**
j=2
to
*k*
**do**7:                  Calculate $C\_W\left(S_{i},CH_{j}\right)$8:                  **if**
$C\_W\left(S_{i},CH_{j}\right)>temp$
**then**9:                      $temp=C\_W\left(S_{i},CH_{j}\right)$10:                    $S\_clusterhead\left(i\right)=j$11:                 **end if**12:          **end for**13:      **end if**14: **end for**

## 6. Proposed Approach

Our proposed approach consists of two phases, i.e., area coverage optimization and energy consumption optimization. To achieve high coverage ratio, low redundancy ratio and energy consumption balance, we utilize an improved adaptive particle swarm optimization (IAPSO) to solve multi-objective area coverage optimization model and cluster head selection optimization model.

### 6.1. Improved Adaptive Particle Swarm Optimization

The particle swarm optimization (PSO) is an evolutionary optimization technology by choosing a number of particles to search for best solution [41]. In PSO, each particle represents a random solution for a specific problem, and it can calculate the fitness function value. The position and velocity of the particle is given by a directional vector respectively. During the process of iteration, each particle search for a better solution dynamically by personal best value $P_{besti}$ and global best value $G_{besti}$. Let there are *m* particles in *h*-dimensional space, the position of each particle $i\left(1\leq i\leq n\right)$ during *t*th iteration is $X_{ih}\left(t\right)=\left(x_{i1},x_{i2},\ldots ,x_{ih}\right)$, and the velocity is $V_{ih}\left(t\right)=\left(v_{i1},v_{i2},\ldots ,v_{ih}\right)$. The best position a particle has experienced is $P_{besti}=\left(p_{i1},p_{i2},\ldots ,p_{ih}\right)$, and the best position all the particles has experienced is $G_{best}=\left(g_{i1},g_{i2},\ldots ,g_{ih}\right)$. During each iteration, each particle updates its velocity and position according to the following equations.
(18)$$V_{ih}\left(t+1\right)=\omega \cdot V_{ih}\left(t\right)+c_{1}\cdot rand_{1}\cdot \left(P_{besti}-X_{ih}\left(t\right)\right)+c_{2}\cdot rand_{2}\cdot \left(G_{besti}-X_{ih}\left(t\right)\right)$$
(19)$$X_{ih}\left(t+1\right)=X_{ih}\left(t\right)+V_{ih}\left(t+1\right),1\leq i\leq n,1\leq h\leq H$$where ω,0<ω<1 is the inertia weight, $c_{1},c_{2},0\leq c_{1},c_{2}\leq 2$ are the acceleration coefficients and $rand_{1},rand_{2},0\leq rand_{1},rand_{2}\leq 2$ are random values. To achieve better optimization, this paper make an improvement for inertia weight.

In PSO, the inertia weight ω play a vital role in adjusting velocity by considering how the former velocity affect current velocity. To adapt to actual situations, it is necessary to adjust the pace of searching for best solution by improving inertia weight. The bigger inertia weight ω benefits a global search, and the smaller one benefits a local search. The traditional linear decrement inertia weight is as follows.
(20)$$\omega =\omega_{\max}-\frac{\omega_{\max}-\omega_{\min}}{T_{\max}}\times t$$where $T_{\max}$ is the maximal number of iteration, and *t* is the current number of iteration. $\omega_{\max}$ is the maximal inertia weight, and $\omega_{\min}$ is the minimal inertia weight. Although the inertia weight can effectively avoid the fluctuation of global best position, it cannot be guaranteed to avoid being caught in local optima with the number of iteration increasing. To solve the problem of partial best, it is necessary to balance the global and partial search. The inertia weight in this paper is defined by the following equation.
(21)$$\omega \left(k\right)=\left\{\begin{array}{c} \omega_{\max}-\frac{\omega_{\max}-\frac{1}{2}\omega_{\max}}{\frac{2}{5}it_{\max}}\times k,k\leq \frac{2}{5}it_{\max}\\ \left(\frac{1}{2}\omega_{\max}-\omega_{\min}\right)\times e^{-\beta \cdot \left(k-\frac{2}{5}it_{\max}\right)}+\omega_{\min},k>\frac{2}{5}it_{\max} \end{array}\right.$$where the inertia weight is a non-linear dynamic adaptive value. $it_{\max}$ is the maximal number of iteration, and *k* is the current number of iteration. $\omega_{\max}$ is the maximal interation weight, and $\omega_{\min}$ is the minimal weight. When the number of iteration *k* is in the interval $\left[0,\frac{2}{5}it_{\max}\right]$, the linear decreasing of the inertia weight benefit to getting close to global optima and avoid local optima. After *k* is over $\left[0,\frac{2}{5}it_{\max}\right]$, the index decrease of inertia weight benifit to accurate search and getting close to optimal area quickly.

### 6.2. Multi-Objective Area Coverage Optimization Based on IAPSO

Let there are *n* directional sensor nodes randomly deployed in the monitoring area, and the position of them are fixed. To improve coverage quality, the improved adaptive particle swarm optimization (IAPSO) algorithm is utilized to improve coverage ratio and reduce redundancy ratio by solving a multi-objective area coverage optimization model. In each iteration, the position and velocity update using Equations (Equation 18) and (Equation 19). The pseudo code of the proposed multi-objective area coverage optimization algorithm based on IAPSO (IAPSO-MOACO) is shown in Algorithm 3.

Algorithm 3 shows the process of coverage optimization by iteration. Firstly, randomly deployed sensor nodes group, predefined swarm size, number of dimensions of particles, and largest number of iterations are initialized. Then, the fitness function values of coverage optimization are calculated by using Equation (Equation 12), and the personal best and global best are derived. With the particle constantly search for the better personal best, the fitness function values gradually tend to be best. Finally, the sensing direction groups are derived.

**Algorithm 3:** Multi-objective area coverage optimization algorithm based on IAPSO**Input:** Sensor nodes group: $s=\left\{s_{1},s_{2},\ldots ,s_{n}\right\}$   Predefined swarm size: num   Number of dimensions of particles: $D_{1}=n$   Largest number of iterations: *maxnumber***Output:** Coverage ratio and redundancy ratio: *CoverageRatio*, *RedundancyRatio*1: Initialize particle $P_{1i},\forall i,j,1\leq i\leq num,1\leq j\leq D_{1}=n$, $X_{1i,j}\left(0\right)=\left(x_{1i,j}\left(0\right),y_{1i,j}\left(0\right)\right)$2: **for**
i=1
to
num
**do**3:       (1) Calculate $Fitness1\left(P_{1i}\right)$, Using Equation (Equation 12)4:       (2) $P_{1besti}=P_{1i}$5: **end for**6: $G_{1best}=\left\{P_{1besti}|Fitness1\left(P_{1besti}\right)=\min\left(Fitness\left(P_{1besti}\right),\forall i,1\leq i\leq num\right)\right\}$7: **for**
t=1
to
maxnumber
**do**8:       **for**
i=1
to
num
**do**9:             (1) Update velocity and position of $P_{1i}$ using Equations (Equation 18) and (Equation 19)10:           (2) Calculate $Fitness1\left(P_{1i}\right)$, update $P_{1besti}$ and $G_{1besti}$11:           (3) Calculate the coverage ratio and redundancy ratio using Equations (Equation 10) and (Equation 11)12:      **end for**13: **end for**

### 6.3. Cluster Head Selection Optimization Based on IAPSO

To achieve energy consumption balance, the improved adaptive particle swarm optimization (IAPSO) algorithm is utilized to select optimal cluster head group by solving cluster head selection optimization model. In each iteration, the position and velocity update using Equations (Equation 18) and (Equation 19). The pseudo code of the proposed cluster head selection optimization algorithm based on IAPSO (IAPSO-CHSO) is shown in Algorithm 4.

**Algorithm 4:** Cluster head selection optimization algorithm based on IAPSO**Input:** Sensor nodes group: $s=\left\{s_{1},s_{2},\ldots ,s_{n}\right\}$   Predefined swarm size: num   Number of dimensions of particles: $D_{2}=k$   Largest number of iterations: *maxnumber***Output:** Cluster head group: $CH=\left\{CH_{1},CH_{2},\ldots ,CH_{k}\right\}$1: Initialize particle $P_{2i},\forall i,j,1\leq i\leq num,1\leq j\leq D_{2}=k$, $X_{2i,j}\left(0\right)=\left(x_{2i,j}\left(0\right),y_{2i,j}\left(0\right)\right)$2: **for**
i=1
to
num
**do**3:       (1) Calculate $Fitness2\left(P_{2i}\right)$, Using Equation (Equation 15)4:       (2) $P_{2besti}=P_{2i}$5: **end for**6: $G_{2best}=\left\{P_{2besti}|Fitness2\left(P_{2besti}\right)=\min\left(Fitness2\left(P_{2besti}\right),\forall i,1\leq i\leq num\right)\right\}$7: **for**
t=1
to
maxnumber
**do**8:       **for**
i=1
to
num
**do**9:             (1) Update velocity and position of $P_{2i}$ using Equations (Equation 18) and (Equation 19)10:           (2) Calculate $Fitness2\left(P_{2i}\right)$, update $P_{2besti}$ and $G_{2besti}$11:           (3) Output the optimal cluster heads group $G_{2besti}$12:      **end for**13: **end for**

Algorithm 4 shows the process of cluster head selection optimization by iteration. Firstly, randomly deployed sensor nodes group, predefined swarm size, number of dimensions of particles, and largest number of iterations are initialized. Then the fitness function values of cluster head selection optimization are calculated by using Equation (Equation 15), and the personal best and global best are derived. With the particle constantly search for the better personal best, the fitness function values gradually tend to be best. Finally, cluster head group are derived.

## 7. Performance Evaluation

### 7.1. Simulation Environment

In this section, we conducted simulation experiments to verify the efficiency and the stability of the proposed approach in MATLAB (version 7.11) and on an Intel core i7 processor with chipset 2600, 3.40 GHZ CPU 4GB RAM running on the platform Microsoft Window 10. In simulation enviroments, the number of directional sensor nodes vary from 100 to 200, and the cluster head ratio is set as 6%. The network parameter and IAPSO parameter are shown in Table 1 and Table 2 respectively.

### 7.2. Performance Evaluation

In the process of simulation, we compare the proposed coverage and energy consumption optimization approach with existing ones. In terms of coverage optimization, the proposed approach is compared with simulated anneal (SA) and random deployment (RD) under the circumstance that cluster-based energy efficiency approach is the same. In terms of energy consumption optimization, the approach is compared with LEACH, LEACH-C, PSO-C. To evaluate the performance of our proposed approach, we consider the four factors, namely coverage ratio, redundancy ratio, number of alive nodes and number of data packets received by BS.

#### 7.2.1. Comparison of Coverage Ratio

In this experiment we run the approaches for comparing the coverage ratio of DSN by varying number of sensor nodes from 100 to 200 and number of cluster heads from 6 to 12. The performance of IAPSO-MOACO is compared with simulated anneal (SA) and random deployment (RD) under the circumstance the cluster-based energy efficiency approach is the one proposed in this paper. The comparison results of the IAPSO-MOACO with other approaches are shown in Figure 4a–c.

As can be seen from the figures, the IAPSO-MOACO has better performance than SA and RD in most of time because the proposed IAPSO algorithm can effectively avoid local optima and improve coverage ratio. With the round of data transmission increase, the coverage ratio will gradually decrease because the number of alive nodes decreases. However, the IAPSO-MOACO can still guarantee higher covergae ratio in comparison with other approaches when the round is no more than 700. It also can be noted that the IAPSO-MOACO still has better optimization performance as the size of the network increases.

#### 7.2.2. Comparison of Redundancy Ratio

In this experiment we run the approaches for comparing the redundancy ratio of DSN by varying number of sensor nodes from 100 to 200 and number of cluster heads from 6 to 12. The performance of the IAPSO-MOACO is compared with SA and RD under the circumstance the cluster-based approach is the same one proposed in this paper. The comparison results of the IAPSO-MOACO with other approaches are shown in Figure 5a–c.

It can be observed from the figures that the IAPSO-MOACO outperformances the SA and RD in most of time because the proposed IAPSO can effectively avoid local optima and reduce the redundancy ratio. As the round of data transmission increases, the redundancy ratio gradually decrease with the number of alive nodes gradually decreases. However, the IAPSO-MOACO still has better optimization performance when the round is no more than 700. It also can be seen that the IAPSO-MOACO can still guarantee lower redundancy ratio in comparison with other approaches with the size of the network changes.

#### 7.2.3. Comparison of Number of Alive Nodes

In this experiment we run the approaches for comparing the number of alive nodes by varying number of sensor nodes from 100 to 200 and number of cluster heads from 6 to 12. The number of alive nodes in each round will directly reflect whether the optimization approach can effectively guarantee energy consumption balance. The comparison results of IAPSO-CHSO with LEACH, LEACH-C and PSO-C are shown in Figure 6a–c.

It can be observed from the three figures that the IAPSO-CHSO can guarantee higher number of alive nodes because the approach not only considers the residual energy of cluster head candidates but also consider the energy consumption balance of them in a round. In contrast, LEACH and LEACH-C do not consider the residual energy when selecting cluster head. The proper cluster head selction optimization model makes the energy consumption evenly distribute on each node which has higher residual energy. The PSO-C consider the residual energy of cluster head, but cannot guarantee the energy consumption balance of cluster heads. As the size of the network increases, the IAPSO-CHSO still has better performance in comparison with other approaches, and 70% sensor nodes was alive during the early 700 rounds under different circumstances.

#### 7.2.4. Comparison of Number of Data Packets Received by BS

In this experiment, we run the approaches for comparing the number of data packets received by BS with varying number of sensor nodes from 100 to 200 and cluster head from 6 to 12. The number of data packets received by BS is directly affected by the number of alive nodes. The data packets received by the base station with different number of initial sensor nodes are shown in Figure 7a–c.

It can be observed from the three figures that the IAPSO-CHSO has advantage over LEACH, LEACH-C and PSO-C in terms of data packets received by BS. The reason is that our proposed approach can guarantee higher number of alive nodes by achieving energy consumption balance. The IAPSO-CHSO has much higher number of packets receipt in comparison with existing approaches after the round is over 500. As the round of data transmission increases, the IAPSO-CHSO approach accomplish the cluster head selection by using proper energy efficiency fitness function. It also can be noted from the figures that the performance of IAPSO-CHSO approach is still better than existing ones with the size of the network increases.

To sum up, the simulation results show that our proposed approach can effectively achieve high coverage ratio, low redundancy ratio and energy consumption balance. For area coverage optimization, we can guarantee higher coverage ratio and lower redundancy ratio compared to SA and RD. For energy consumption optimization, we can guarantee energy consumption balance compared to LEACH, LEACH-C and PSO-C.

## 8. Conclusions

In this paper, we proposed an area coverage and energy consumption optimization problem based on IAPSO for DSN. First, we set up a multi-objective area coverage optimization model in order to improve coverage ratio and reduce redundancy ratio. Then, we set up an energy efficiency cluster head selection optimization model and proposed a cluster formation algorithm based on weight function. We ultilized IAPSO-MOACO to achive high coverage ratio, low redundancy ratio by sensing direction rotation. We also ultilized IAPSO-CHSO to achieve energy consumption balance by reasonable cluster head selection.

We conducted simulation experiments to demonstrate the advantages of our proposed approach. For area coverage optimization, the comparison with existing approaches showed that our proposed approach could effectively improve coverage ratio and reduce redundancy ratio when the round of data transmission was no more than 700. The IAPSO-MOACO had better optimization performance in most of time because the IAPSO could effectively avoid local optima to some degree. For energy consumption optimization, the comparison with existing approach showed that the IAPSO-CHSO could effectively achieve energy consumption balance because it could guarantee higher number of alive nodes and data packets received by BS. During the early 700 rounds, 70% sensor nodes was still alive when we ran the IAPSO-CHSO in networks with different sizes. The number of data packets received by BS was much higher than other approches after the round was over 500. The proposed approach could effectively guarantee energy consumption balance.

## Figures and Tables

**Figure 1 sensors-19-01192-f001:**
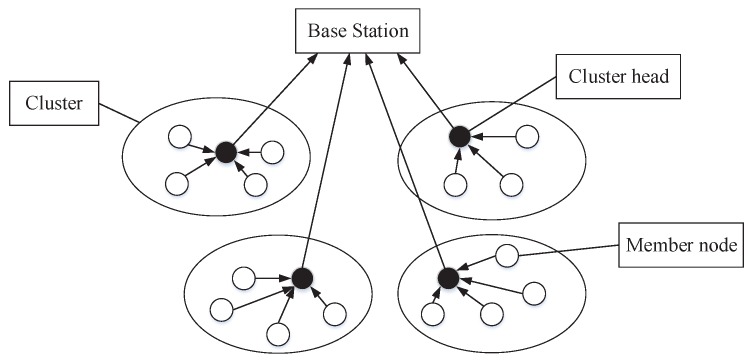
Cluster based directional sensor network.

**Figure 2 sensors-19-01192-f002:**
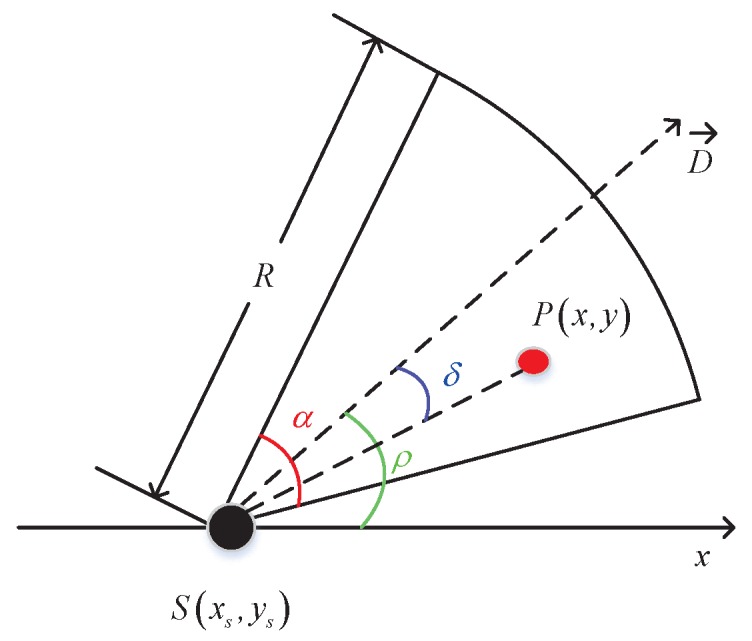
Directional sensor model.

**Figure 3 sensors-19-01192-f003:**
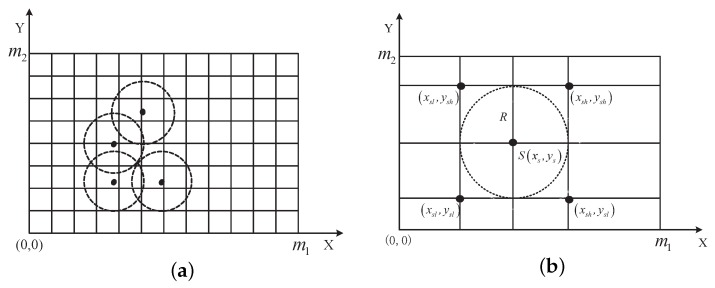
Coverage situations varication of monitoring area. (**a**) Monitoring area gridding; (**b**) Sensing range of sensor node.

**Figure 4 sensors-19-01192-f004:**
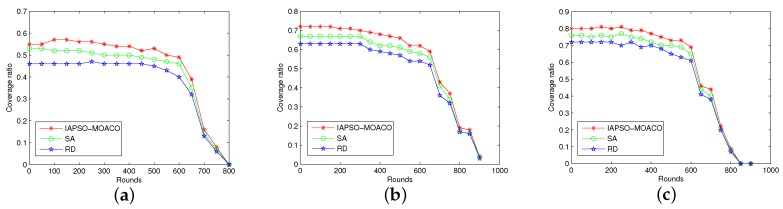
Comparison in terms of coverage ratio with 100 to 200 nodes. (**a**) 100 nodes with 6 cluster heads; (**b**) 150 nodes with 9 cluster heads; (**c**) 200 nodes with 12 cluster heads.

**Figure 5 sensors-19-01192-f005:**
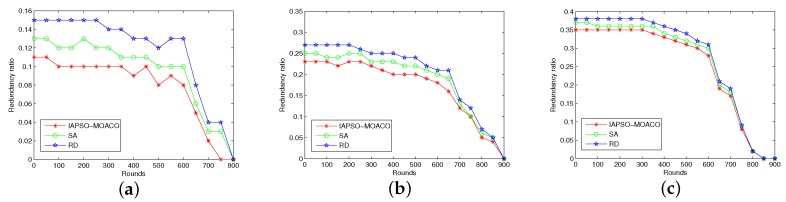
Comparison in terms of redundancy ratio with 100 to 200 nodes. (**a**) 100 nodes with 6 cluster heads; (**b**) 150 nodes with 9 cluster heads; (**c**) 200 nodes with 12 cluster heads.

**Figure 6 sensors-19-01192-f006:**
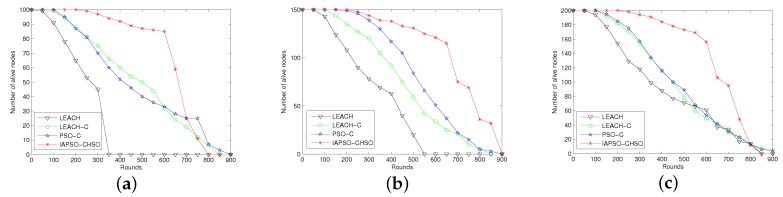
Comparison in terms of number of alive nodes with 100 to 200 nodes. (**a**) 100 nodes with 6 cluster heads; (**b**) 150 nodes with 9 cluster heads; (**c**) 200 nodes with 12 cluster heads.

**Figure 7 sensors-19-01192-f007:**
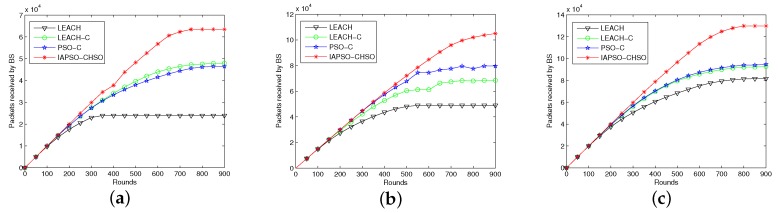
Comparison in terms of data packets received by BS with 100 to 200 nodes. (**a**) 100 nodes with 6 cluster heads; (**b**) 150 nodes with 9 cluster heads; (**c**) 200 nodes with 12 cluster heads.

**Table 1 sensors-19-01192-t001:** Network parameters.

Parameter	Value
Target area	500 × 500 m${}^{2}$
Base Station position	(250,250)
Number of directional sensor nodes	100–200
Number of cluster heads	6–12
Sensing radius of directional sensor nodes	6 m
Sensing angle of directional sensor nodes	π/6
Intitial energy of directional sensor nodes	2 J
$E_{elec}$	50 nj/bit
$\varepsilon_{fs}$	10 pJ/bit/m${}^{2}$
$\varepsilon_{mp}$	0.0013 pJ/bit/m${}^{4}$
$d_{0}$	87 m
Packet length	4000 bits
Message size	500 bits

**Table 2 sensors-19-01192-t002:** IAPSO parameters.

Parameter	Value
Number of particles	30
$C_{1}$	2
$C_{2}$	2
ω	[0,1]
$D_{1}$	[0,200]
$D_{2}$	[6,12]
$V_{\max}$	2π
Number of iterations	500

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
