# Peer review of "An Area Coverage and Energy Consumption Optimization Approach Based on Improved Adaptive Particle Swarm Optimization for Directional Sensor Networks"

_sensors, 2019, doi:10.3390/s19051192_

Round 1
Reviewer 1 Report
This paper considers two problems. One is the area coverage optimization problem in directional sensor networks. The authors consider rotating the sensing direction to reduce the coverage of blind and overlapping areas. The second problem is the energy consumption optimization problem in general wireless sensor networks. The authors consider selecting cluster heads to achieve communication energy consumption balance. The authors then propose using the improved adaptive particle swarm optimization (IAPSO) algorithm to solve the two problems individually.
It seems unnecessary and weird to put the two problems in one paper because the second one is not a unique problem to directional sensor networks.
To optimize the area coverage of directional sensor networks, why not directly optimize the area coverage, the second term in Eqn. 10?
Sections 4 and 5 contain too many trivial details. The paper could be seriously shortened.
The proposed IAPSO is a heuristic solution. The paper needs to give a fair comparison with existing solutions.
Author Response
Dear Reviewer, we have revised the paper according to your advice. There are a revised version of paper and a response letter. We have marked in the revised content.
Best Regards!

Reviewer 2 Report
The development of the utilized mechanisms is well done.The major issue I noticed is the description of the results and their discussion. The authors provide randomized simulation results, but lack a description as of to the details thereof (placements etc) and the statistical significant of their results (e.g., confidence intervals).
Author Response

(The authors gave the same response as above.)

Reviewer 3 Report
The paper tried to propose a novel area coverage and energy consumption optimization approach based on improved adaptive particle swarm optimization (IAPSO). The authors clearly presented the three key contributions with the performance evaluation including coverage ratio, redundancy ratio and energy consumption balance in the form of the number of alive node and the number of data packets received by BS. The small change should be added and clarified.
The paper is well written, clearly presented the proposed techniques with the evaluation.
However, the author should explain more a bit for the following comments:
Why selecting particle swarm optimization over another optimization technique?
The paper focus only on the communication energy for data transmission, but not for the energy consumption for data collection. Why it shouldn’t be concerned particular for the system using solar energy?
It should have more detail description for the result and conclusion.
Are there any example application or possible implementation of sensor network by using this technique? Sample scenario would be fine
Author Response

(The authors gave the same response as above.)
